# OJBench: A Competition Level Code Benchmark For Large Language Models

## Abstract

Recent advancements in large language models (LLMs) have demonstrated remarkable progress in mathematical and coding reasoning. However, existing code benchmarks are limited in their ability to evaluate the full spectrum of these capabilities, especially at the level of top-tier human programming competitions. To bridge this gap, we introduce **OJBench**, a novel and challenging benchmark designed to assess the competitive-level code reasoning abilities of LLMs. OJBench comprises 232 programming competition problems from NOI and ICPC, providing a rigorous test of models' reasoning skills. We conducted a comprehensive evaluation of 37 models on OJBench, including a mix of closed-source, open-source, reasoning-oriented, and general-purpose models. Our results indicate that even state-of-the-art reasoning models like o4-mini and Gemini-2.5-pro-exp struggle with highly challenging, competition-level problems, highlighting the significant challenges models face in this domain.

## 1 Introduction

With the emergence of long chain-of-thought (CoT) models such as OpenAI-o1, o4-mini (Jaech et al., 2024), Gemini-2.5-pro-ex (Google, 2025), DeepSeek-R1 (Guo et al., 2025) and Qwen3 (Team, 2025a), large language models have demonstrated advanced reasoning capabilities across various domains, including mathematics (Hendrycks et al., 2021), physics (Welbl et al., 2017), and formal proof (Wang et al., 2025; DeepSeek-Prover, 2025). They have also shown significant progress in applications such as code generation and software engineering (Fan et al., 2023). Math and code are two domains commonly used to evaluate the reasoning abilities of models. Although the field of mathematics has numerous competition-level benchmarks, such as Omni-Math (Gao et al., 2024), and AIME, the field of code still lacks competition-level benchmarks that can pose a challenge to models. Currently, researchers primarily evaluate models' performance on complex algorithmic programming tasks using LiveCodeBench (Jain et al., 2024). However, the scope of LiveCodeBench problems is often limited, as they primarily evaluate a single knowledge point and can be addressed with minimal coding effort. The observed performance saturation among various models on Live-CodeBench suggests that the current benchmark is insufficient for distinguishing their capabilities. This underscores the necessity for more complex benchmarks to reveal model limitations and inform the development of more advanced code LLMs.

An ideal scenario is to use real competitive programming tasks, which are often designed for top-tier programmers selected globally. Inspired by the practice of testing models' coding abilities on the CodeForces platform by models such as OpenAI-o1, o3 (openaiteam, 2024), and DeepSeek-R1 (Guo et al., 2025), the recent study CodeElo (Quan et al., 2025) has explored how to evaluate the competitive coding abilities of different models. However, this approach utilizes simulated CodeForces submissions and lacks a standardized, transparent dataset. Consequently, the choice of problems can introduce a significant bias into a model's Elo rating, which hinders a robust evaluation of its competition-level coding capabilities.

To bridge this gap, we propose OJBench, a competition-level code benchmark comprising 232 problems that can effectively assess models' code reasoning abilities in competitive programming tasks. Using OJBench, we evaluated 37 models, including the Coder models trained on large-scale code corpora and reasoning-oriented models trained with reinforcement learning on a wide range of reasoning tasks. We found that reasoning-oriented models significantly outperformed general-purpose

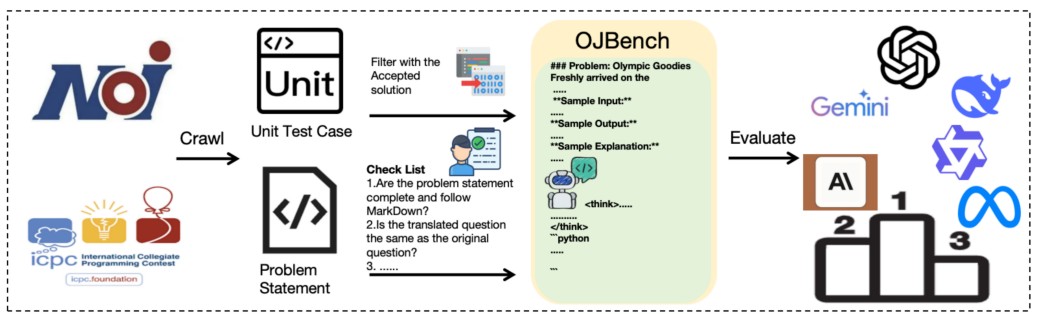

Figure 1: Overall data collection and filtering process of OJBench.

models in competitive coding tasks, and open-source models still lag behind closed-source models in terms of code reasoning proficiency. Additionally, we found that for most long-CoT models, using CPP proves more effective than Python. Lastly, these models can leverage execution environment feedback to iteratively refine their solutions, which ultimately enhances their overall performance.

In summary, our main contributions are as follows:

1) We introduce OJBench, a competition-level code benchmark that encompasses 232 top-tier human programming competitions.

2) Utilizing OJBench, we evaluated 37 models and revealed the limitations of current models in complex code reasoning tasks.

3) We evaluated how well large language models code in different programming languages in complex programming tasks and examined their ability to self-correct buggy solutions by leveraging environmental feedback. Our findings offer actionable insights for the future development of code-centric LLMs.

## 2 OJBENCH

In Section 2.1, we introduced the sources of the problems in OJBench and the data collection process. In Section 2.2, we demonstrated how we assigned difficulty labels to each problem. In Sections 2.3 and 2.4, we respectively elaborated on the evaluation methods employed by OJBench and its support for dual-language assessment in both Python and CPP.

### 2.1 DATA SOURCE AND DATA COLLECTION

The overall data collection and data filtering process of OJBench is shown in Figure 1.

**Data Source.** First we reviewed existing benchmarks, such as USACO (Shi et al., 2024), contain problems exclusively from the USA Computing Olympiad. LiveCodeBench (Jain et al., 2024) and codeELO (Quan et al., 2025), on the other hand, utilize problems from the LeetCode and Codeforces platforms respectively. To minimize overlap with existing benchmarks and incorporate more challenging competitive programming problems into our dataset, we selected problems from China's National Olympiad in Informatics (NOI) and International Collegiate Programming Contest (ICPC) as our data sources. For detailed information on NOI and ICPC, please refer to Appendix H.

**Data Collection.** We collected NOI and ICPC questions from the Logu competition platform and the ICPC official website respectively. Each data sample includes problem descriptions in Markdown format and the comprehensive test cases released by the competition organizers. These extensive test cases provides a solid foundation for our evaluation set, thereby ensuring its validity. Subsequently, we conducted a simple data filtering process:

1) All collected test cases were validated against accepted contestant submissions; any sample that failed the automated verdict was discarded, eliminating artifacts arising from incomplete or incorrect test suites.

Table 1: Statistics of problems collected in OJBench

| Source | Count | Easy | Medium | Hard | Average Tests |
|--------|-------|------|--------|------|---------------|
| NOI | 159 | 20 | 53 | 86 | 17.21 |
| ICPC | 73 | 16 | 26 | 31 | 63.60 |
| Total | 232 | 36 | 79 | 117 | 31.81 |

2) The original dataset contained tasks whose outputs are non-unique and whose correctness can only be determined by bespoke special judges. Due to the frequent absence and non-trivial authoring cost of such graders, we excluded every task that demands a custom judge.

Finally, we used GPT-4o to translate the problem descriptions of NOI into English, followed by manual verification to ensure that the translated problems retained their original meaning and maintained the correct format. The statistics of OJBench are presented in Table 1.

## 2.2 DIFFICULTY CLASSIFICATION

To enable a fine-grained assessment of the capability of the model, we stratified the OJBench tasks into three ordinal difficulty tiers: Easy, Medium, and Hard.

**NOI.** For problems sourced from NOI, we utilized the difficulty ratings obtained from the competition platform for categorization. Specifically, the difficulty of each problem is annotated through voting by contestants who successfully solved the problem. The difficulty of each problem is quantified on a scale from 0 to 7, with higher values indicating greater difficulty. The difficulty range of OJBench problems spans from 2 to 7. We categorized problems with a difficulty level of 2-3 as Easy, 4-5 as Medium, and 6–7 as Hard.

**ICPC.** For problems sourced from ICPC, due to the lack of an official difficulty classification, we crawled the ranking information from real competitions. For each problem, we considered four metrics: total submission count (total_submission), total number of accepted solutions (total_passed), total number of participating teams (total_team), and the number of teams that attempted the problem (attempted_team). We calculated the difficulty score for each problem using the following formula:

$$\text{score} = \left( \frac{\text{total\_passed}}{\text{total\_submission}} \right) \times \left( \frac{\text{attempted\_team}}{\text{total\_team}} \right)$$

The ratio $\frac{\text{total\_passed}}{\text{total\_submission}}$ measures the success rate of teams attempting to solve the problem. In real programming contests, teams often assess the difficulty of a problem before deciding whether to attempt it. Therefore, the ratio $\frac{\text{attempted\_team}}{\text{total\_team}}$ indicates the perceived difficulty level of the problem by the participating teams. This score effectively captures the true difficulty of the problem. Specifically, we classified problems with a score of 0.4 or higher as Easy, those with a score of 0.1 or lower as Hard, and the rest problems as Medium.

## 2.3 EVALUATION METHOD

**Judge Base on the test case.** In programming competitions, test cases are commonly used to verify the correctness of solution. In programming tasks, models are provided with problem descriptions and input-output examples, and are required to generate correct solutions. The correctness of these solutions is evaluated based on a set of test cases (input-output pairs). Previous research has indicated that using a limited number of test cases for evaluation can easily lead to false positives results (Yang et al., 2025). Similarly, in our experiments, we observed that using a small number of test cases for validation tends to produce false-positive results. As shown in Figure 2, the performance of all models decreases with an increasing number of test cases. This trend persists even when evaluated on powerful, reasoning-oriented models. To rigorously validate the robustness of the code, we assess the correctness of the programs generated by the models on the entire set of test cases. A solution is considered correct only if it passes all test cases.

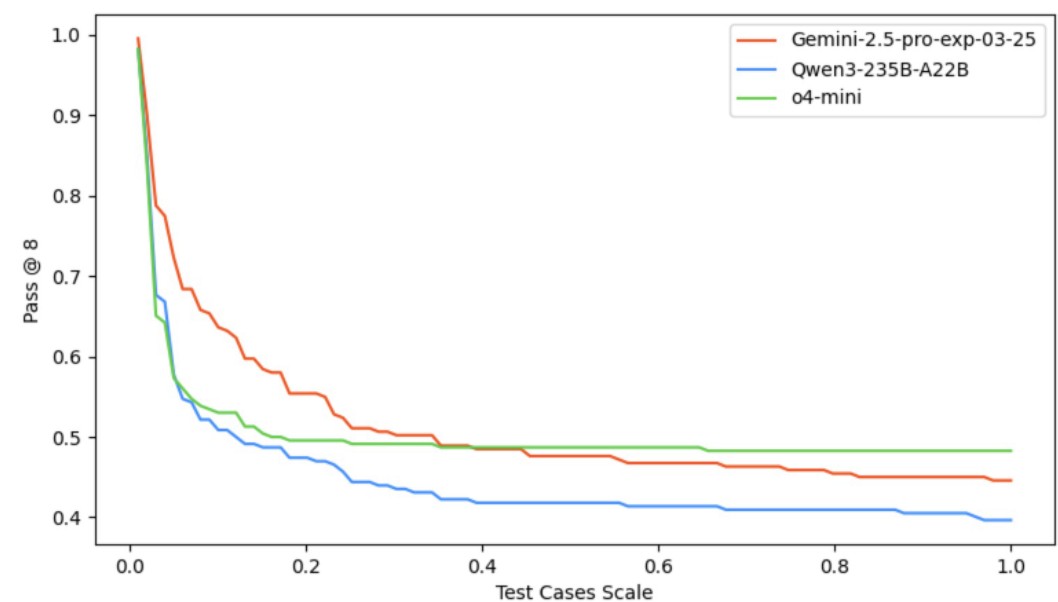

Figure 2: Performance decline of models with increasing test case scale on OJBench.

## 2.4 DUAL PROGRAMMING LANGUAGE ASSESSMENT.

Existing code benchmarks, such as LiveCodeBench (Jain et al., 2024) and APPS (Hendrycks et al., 2021), predominantly evaluate models solely on Python. However, competitive programming exhibits a distinct linguistic landscape, where human contestants largely favor CPP over Python, especially for top-tier problems with stringent algorithmic time complexity requirements (Quan et al., 2025). Sole reliance on Python for evaluation thus provides an incomplete assessment of model performance in such contexts. To address this limitation, OJBench supports evaluation in both Python and CPP. This dual-language capability facilitates a nuanced investigation into performance disparities across languages. Our findings indicate that most reasoning-oriented models achieve superior performance when utilizing CPP compared to Python. This observation aligns with human experience, given CPP's inherent efficiency advantages, which make it more suitable for solving competitive-level problems.

## 3 EVALUATION ON EXISTING LLMs

### 3.1 EXPERIMENTAL SETUP

**Metrics.** We employ **Pass@n** (Kulal et al., 2019)as our evaluation metric. Which means that if a model gives n answers to the same problem and at least one of the answers is correct, the model is considered to have passed the problem. During evaluation, we assess the models using the hyper parameters officially recommended by each model, including temperature, top_k, and top_p. For reasoning-oriented models, we set the maximum number of tokens (max_tokens) to 64k. For general-purpose models, we set max_tokens to the default context length of the model.

**Models.** Our evaluation includes a diverse range of models. For open-source general-purpose models, we selected models from the Qwen2.5-Coder-Instruct, DeepSeek-Coder-Instruct, and CodeLlama-Instruct series, alongside DeepSeek-V3-0324 (Hui et al., 2024; Zhu et al., 2024; Guo et al., 2024; Liu et al., 2024a; Grattafiori et al., 2024). For reasoning-oriented open-source models, our selection covers the DeepSeek-R1 family (including its distilled variants), various Qwen3 models, QWQ-32B, and the Olympic-Coder series (Team, 2025b; openrl, 2025). Among closed-source models, we assessed leading proprietary offerings, including multiple versions of the Claude-3, Gemini-2, and o-series, as well as models from the GPT line (from GPT-3.5 to GPT-4o) (Hurst et al., 2024; openaiteam, 2024).

Table 2: The main results of different models on OJBench using Python and CPP. Pass@1@Py and Pass@8@Py represent Pass@1 and Pass@8 in Python respectively, and the same is true for CPP. The highest scores of open source general-purpose models, open source reasoning-oriented models, and closed source models are marked in blue, green, and red respectively.

| Model | Pass@ | | | | Pass Rate @ | | |
|---|---|---|---|---|---|---|---|
| | 1@Py | 8@Py | 1@CPP | 8@CPP | Easy | Mid | Hard |
| *General-purpose Open-source LLM* | | | | | | | |
| Qwen2.5-Coder-7B | 3.50 | 4.74 | 2.64 | 4.74 | 20.49 | 0.95 | 0.00 |
| Qwen2.5-Coder-14B | 6.30 | 10.34 | 4.53 | 8.62 | 35.07 | 2.53 | 0.00 |
| Qwen2.5-Coder-32B | 5.77 | 9.05 | 6.36 | 12.93 | 30.56 | 3.01 | 0.00 |
| CodeLlama-7B | 0.00 | 0.00 | 0.00 | 0.00 | 0.00 | 0.00 | 0.00 |
| CodeLlama-13B | 0.59 | 1.29 | 0.86 | 1.29 | 3.82 | 0.00 | 0.00 |
| CodeLlama-34B | 0.11 | 0.86 | 0.48 | 0.86 | 0.69 | 0.00 | 0.00 |
| CodeLlama-70B | 2.53 | 4.74 | 0.92 | 3.02 | 13.54 | 1.27 | 0.00 |
| DeepSeek-Coder-6.7B | 1.99 | 3.88 | 1.19 | 3.45 | 10.07 | 1.27 | 0.00 |
| DeepSeek-Coder-33B | 2.64 | 6.47 | 2.59 | 6.47 | 14.24 | 1.27 | 0.00 |
| DeepSeek-Coder-V2-Lite | 4.80 | 7.76 | 4.04 | 7.76 | 26.04 | 2.22 | 0.00 |
| DeepSeek-Coder-V2 | 8.24 | 11.21 | 8.67 | 14.22 | 44.79 | 3.80 | 0.00 |
| DeepSeek-V3-0324 | 25.54 | 32.33 | 22.95 | 34.05 | 78.47 | 33.70 | 3.74 |
| *Reasoning-oriented Open-source LLM* | | | | | | | |
| QWQ-32B | 19.02 | 30.17 | 19.77 | 31.90 | 69.10 | 20.89 | 2.35 |
| Qwen3-32B | 14.92 | 32.76 | 15.09 | 31.90 | 53.12 | 17.25 | 1.60 |
| Qwen3-30B-A3B | 15.84 | 29.31 | 12.98 | 25.00 | 55.21 | 19.78 | 1.07 |
| Qwen3-235B-A22B | 25.97 | 40.52 | 26.08 | 39.22 | 76.39 | 35.13 | 4.27 |
| Olympic-Coder-7B | 8.78 | 18.10 | 8.24 | 18.97 | 42.36 | 6.17 | 0.21 |
| Olympic-Coder-32B | 15.41 | 29.31 | 16.27 | 31.03 | 54.51 | 18.67 | 1.18 |
| DeepSeek-R1-Distill-Qwen-1.5B | 2.59 | 6.47 | 0.05 | 0.43 | 13.54 | 1.42 | 0.00 |
| DeepSeek-R1-Distill-Qwen-7B | 8.94 | 15.09 | 0.97 | 2.59 | 39.93 | 8.07 | 0.00 |
| DeepSeek-R1-Distill-Qwen-14B | 14.17 | 23.28 | 5.87 | 13.79 | 56.94 | 14.40 | 0.85 |
| DeepSeek-R1-Distill-Qwen-32B | 17.83 | 30.17 | 10.40 | 24.14 | 62.50 | 19.94 | 2.67 |
| DeepSeek-R1-Distill-Llama-8B | 8.46 | 16.38 | 1.72 | 6.47 | 40.28 | 5.54 | 0.64 |
| DeepSeek-R1-Distill-Llama-70B | 16.38 | 28.02 | 10.02 | 23.71 | 61.81 | 18.20 | 1.18 |
| DeepSeek-R1 | 26.02 | 37.07 | 25.97 | 38.36 | 78.47 | 35.44 | 3.53 |
| *Closed-source LLM* | | | | | | | |
| Claude-3.5-sonnet-20241022 | 10.40 | 17.67 | 13.09 | 23.28 | 47.57 | 8.54 | 0.21 |
| Claude-3.7-sonnet-20250219 | 4.24 | 7.76 | 15.41 | 23.71 | 19.79 | 1.62 | 0.93 |
| Claude-3.7-sonnet-20250219-Thinking | 18.27 | 25.00 | 14.71 | 23.28 | 68.75 | 20.41 | 1.28 |
| GPT3.5-Turbo | 5.60 | 9.05 | 4.26 | 8.62 | 31.25 | 1.74 | 0.32 |
| GPT4-Turbo | 9.97 | 16.81 | 9.70 | 18.53 | 46.18 | 7.28 | 0.64 |
| GPT-4o-20241120 | 10.02 | 15.95 | 10.34 | 16.81 | 50.69 | 5.85 | 0.32 |
| o1-mini | 21.61 | 33.62 | 25.97 | 37.50 | 68.75 | 30.22 | 1.28 |
| o1-20241217 | 26.45 | 35.78 | 33.24 | 47.84 | 81.94 | 35.92 | 2.99 |
| o3-mini | 31.79 | 46.12 | 40.25 | 52.16 | 84.03 | 44.94 | 6.84 |
| o4-mini | 33.30 | 48.71 | 46.12 | 61.21 | 83.33 | 51.27 | 5.77 |
| Gemini-2.0-Flash-Thinking-exp | 13.58 | 22.41 | 11.80 | 20.26 | 54.86 | 13.45 | 0.96 |
| Gemini-2.5-pro-exp-03-25 | 38.91 | 48.71 | 44.26 | 56.47 | 83.68 | 61.71 | 9.48 |

## 3.2 MAIN RESULTS

Table 2 presents the overall performance of all models on OJBench, and the pass@1 scores of the models using Python at different difficulty levels.

**Overall Performance.** In general, closed-source models outperform open-source models. Among open-source reasoning-oriented models, Qwen3-235B-A22B and DeepSeek-R1 demonstrate the best performance, surpassing o1-mini and approaching the performance of o1-20241217. In the closed-source category, previous models such as GPT3.5 and GPT4 showed limitations in competition-level code reasoning tasks. However, models that underwent large-scale post-training specifically for reasoning tasks, such as o4-mini and Gemini-2.5-pro-exp-03-25, exhibit state-of-the-art performance. For models within the same family but of different sizes, performance improves with an increase in parameter count. Overall, models designed for reasoning tasks consistently outperform general-purpose models, highlighting the significant advantages of post-training methods like reinforcement learning and distillation in enhancing reasoning capabilities.

**Pass@n.** As the number of samples n increases, the pass@n of all models consistently rises. Among open-source models, Qwen3-235B-A22B and DeepSeek-R1 exhibit a pass@8 improvement of 14.55 and 11.05 in python, respectively, compared to pass@1, surpassing the performance of o1-20241217 in the pass@8. For closed-source models, pass@8 also shows significant improvement over pass@1. For instance, o3-mini, o4-mini, and Gemini-2.5-pro-exp-03-25 achieve absolute improvements of 14.33, 15.41, and 9.8, respectively. This indicates that models are capable of exploring diverse approaches to problem solving.

**Pass Rate at Different Difficulty Levels.** Table 2 also presents the pass rates of the models implemented in Python across problems of varying difficulty levels in a single response. It is evident that even problems of simple difficulty pose a challenge for the majority of models. For problems at the Hard difficulty level, the pass rates of almost all general-purpose models are zero. Notably, even the DeepSeek-V3-0324 model with an exceptionally large parameter scale of 671B achieved a pass rate of only 3.74%. In contrast, most reasoning-oriented models perform well on problems of easy difficulty. This indicates that merely relying on large-scale pretraining is insufficient to enhance models' performance on competition-level coding problems. Instead, reinforcement learning and distillation from powerful reasoning-oriented models hold significant potential for improving models' code reasoning capabilities. For most reasoning-oriented models, performance on medium and hard problems is relatively lower but provides better differentiation. Hard problems effectively distinguish the top reasoning-oriented models, such as o4-mini, Gemini-2.5-pro-exp-03-25, Qwen3-235B-A22B, and DeepSeek-R1. These results indicate that OJBench can effectively evaluate models' code reasoning abilities and is suitable for assessing the capabilities of future code LLMs.

## 4 ANALYSIS

### 4.1 ANALYSIS OF THE OVERALL DIFFICULTY OF OJBENCH

To provide an objective evaluation of the difficulty inherent in OJBench, a comparative analysis was performed against an established external benchmark. We selected the widely-adopted Live-CodeBench, as its public leaderboard provides performance metrics for state-of-the-art, reasoning-focused models like o4-mini and Gemini-2.5-pro-exp-03-25. Our comparison leverages the officially reported scores for o4-mini (low), Gemini-2.5-pro, and Qwen3-235B-A22B, which were evaluated on the data subset from January 1, 2025, to May 1, 2025.

The comparison results are shown in Table 3. While these three models perform excellently on Live-CodeBench, their significantly reduced performance on OJBench reveals that the overall difficulty distribution of OJBench is demonstrably more demanding than that of LiveCodeBench and indicates that current reasoning models still have significant shortcomings and room for optimization In code reasoning tasks at a competitive level.

### 4.2 COMPARISON BETWEEN CPP AND PYTHON

In addition to Python, we also evaluated the performance of the models using the CPP language on OJBench, the resluts are displayed in Figure 3.

The results indicate that for advanced reasoning-oriented models, such as o4-mini, o1-20241217, and Gemini-2.5-pro-exp-03-25, using CPP as the programming language yields significantly better performance on OJBench compared to Python. We assume that this is due to the fact that

Table 3: Comparison of OJBench and LiveCodeBench based on model performance.

| Model | Benchmark | Pass@1 | Easy-Pass@1 | Mediun-Pass@1 | Hard-Pass@1 |
|---|---|---|---|---|---|
| o4-mini(low) | LCB | 63.70 | 97.80 | 76.40 | 36.60 |
| o4-mini | OJBench | 33.30 | 83.33 | 51.27 | 5.77 |
| Gemini-2.5-pro | LCB | 65.90 | 100 | 74.50 | 41.50 |
| Gemini-2.5-pro-exp-03-25 | OJBench | 38.91 | 83.68 | 61.71 | 9.48 |
| Qwen3-235B-A22B | LCB | 56.60 | 100 | 72.70 | 28.00 |
| Qwen3-235B-A22B | OJBench | 25.97 | 76.39 | 35.13 | 1.07 |

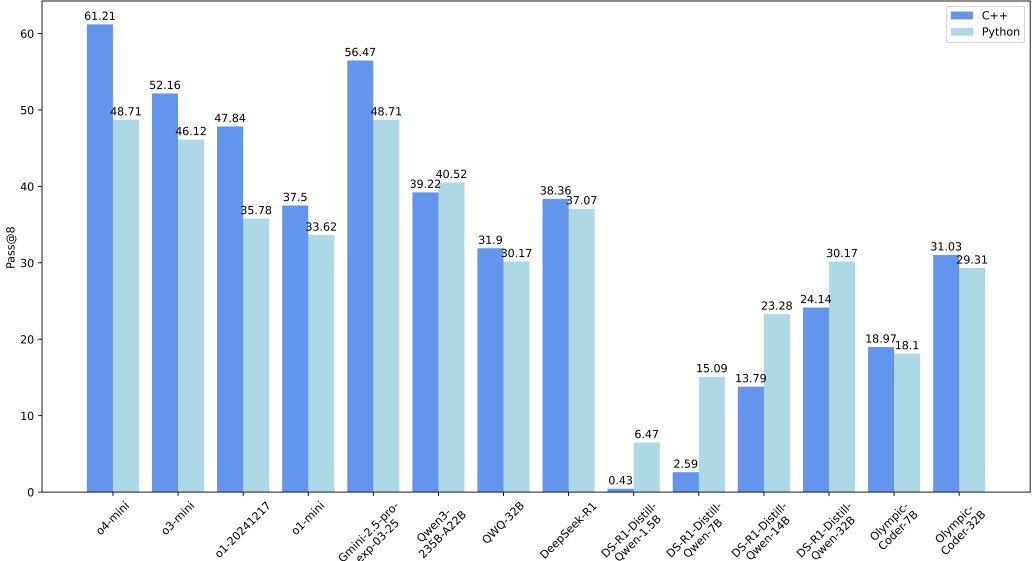

Figure 3: The comparison between CPP and Python.

CPP is inherently a high-performance programming language, making it more suitable for solving competition-level programming tasks than Python.

For the Qwen series models distilled from DeepSeek-R1, the performance of these models using Python is much higher than that of CPP, while the performance of Olympic-Coder-7B/32B using CPP and Python is not much different, with CPP slightly higher than Python. We attribute this to differences in the training data. The former models were trained using a large amount of distilled data from DeepSeek-R1, while the Olympic-Coder-7B/32B models were trained using CPP solutions for competition-level problems from the same teacher model, enabling them to better leverage CPP to solve tasks in OJBench. We provide more details on the pass rates of models using CPP and Python in Appendix F.

### 4.3 REFINEMENT CAN IMPROVE THE PERFORMANCE OF THE MODEL

In real-world competitive programming scenarios, human participants have access to error messages generated during code execution, which they use to debug and correct their code. Motivated by this, we explored whether models could leverage error messages from code execution to rectify erroneous solutions. Specifically, for erroneous model solutions, we directly utilize the solution code and its error feedback (e.g., Compile Error (CE), Time Limit Exceeded (TLE), Wrong Answer (WA)) as prompts to guide subsequent refinement. The results are shown in Figure 4.

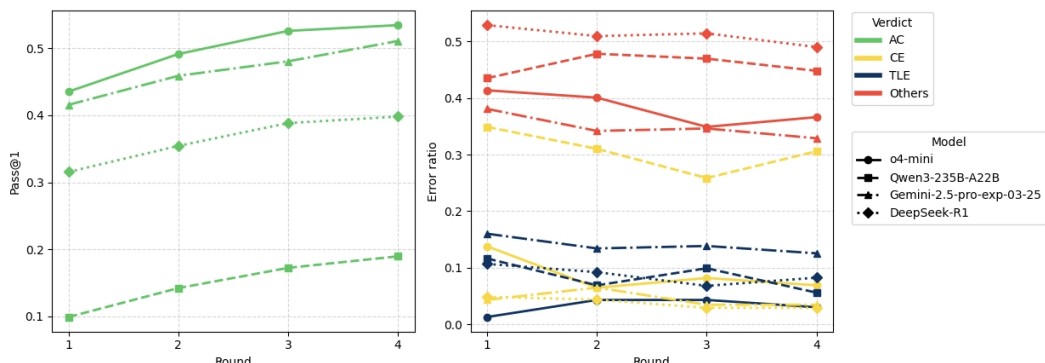

Figure 4: Refinement on OJBench. Among all types, TLE occurs most frequently and CE has no direct correlation with the reasoning ability of the model. Therefore, we distinguish **CE** and **TLE** from other error types **(Others)**. This allows us to more clearly understand which error types the model can reduce through refinement.

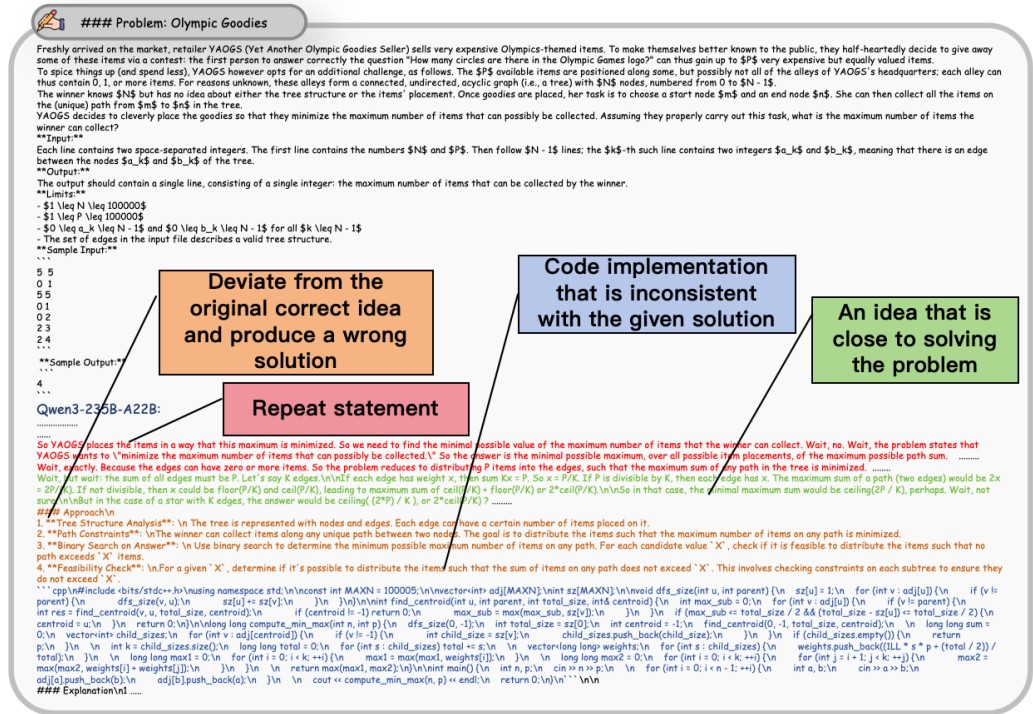

Figure 5: Case analysis on OJBench

We observed that through each round of refinement, the model can continuously improve the pass rate of the solution. Among different types of errors, the proportion of CE errors decreased most significantly. We assume that this is because CE are not directly related to the model's ability to solve competitive programming tasks but rather stem from the model's oversight of certain details during the coding process. Therefore, they can be easily corrected using the error messages.

However, we found that the model struggled to address TLE errors during the refinement process. This is because resolving TLE requires the model to design more efficient algorithms and data structures tailored to specific programming tasks, which is essential for solving complex competitive programming problems. This shows that models still face significant challenges in designing efficient algorithms to solve complex code reasoning tasks.

## 4.4 ANALYSIS OF PROBLEMS THAT THE MODEL CANNOT SOLVE

To thoroughly investigate the problem-solving strategies of large language models with reasoning capabilities, we selected Qwen3-235B-A22B for case analysis. This selection was made due to the inaccessibility of the internal reasoning processes of closed-source models. Figure 5 presents a detailed case analysis of the reasoning process of Qwen3-235B-A22B.

In this case, the model exhibited a significant amount of repetitive restatements of the problem requirements during the reasoning process, rather than engaging in a deeper analysis. Subsequently, the model developed a solution approach that was close to the correct line of reasoning. However, it failed to accurately assess the feasibility of this approach in subsequent steps, ultimately leading to an incorrect problem-solving strategy. Moreover, the code implemented by the model did not align with the proposed solution approach.

## 5 RELATED WORK

**Code Large Language Models.** In recent years, numerous large language models specifically designed for code-related tasks have emerged, such as AlphaCode (Li et al., 2022), StarCoder (Li et al., 2023), and Qwen2.5-Coder (Hui et al., 2024). These models, trained on extensive code corpora, have demonstrated remarkable capabilities in tasks related to code generation (Yu et al., 2024), completion (Chai et al., 2024), and debugging (Liu et al., 2024b). However, while these code LLMs have achieved significant improvements in simple code tasks, such as code completion and repair, they exhibit limited reasoning abilities when dealing with complex programming problems.

**Reasoning-oriented Large Language Models.** Recently reasoning-oriented Large Language Models have demonstrated formidable reasoning capabilities across various domains (Team et al., 2025; Liu et al., 2025), particularly those trained with large-scale reinforcement learning, such as OpenAI-o1, o3 (openaiteam, 2024), DeepSeek-R1 (Guo et al., 2025). These models have shown reasoning abilities comparable to those of human competitive programmers in complex tasks involving mathematics and coding, rendering existing evaluation benchmarks insufficient for accurately assessing model performance. This makes it more urgent to develop competition-level evaluation datasets.

**Code Generation Benchmarks.** Previous code generation benchmarks, such as code generation benchmarks, such as HumanEval (Chen et al., 2021), DS-1000 (Lai et al., 2023), have primarily focused on evaluating models' abilities to generate simple functions, perform data operations. However, these benchmarks fail to provide a discriminative assessment for advanced Reasoning Large Language Models. Additionally, there are benchmarks specifically designed to evaluate models' performance on complex code generation problems, such as CodeELO (Quan et al., 2025), Probench (Yang et al., 2025). These benchmarks are more complex than their predecessors and place higher demands on models' reasoning abilities. Nonetheless, the private test cases and opaque datasets render them unsuitable for use as LLM benchmarks. OJBench, focusing on premier human programming competitions, fills a critical gap by providing demanding code-generation tasks for contemporary reasoning models. It thus holds the potential to serve as a unified and equitable benchmark, informing the advancement of code LLMs.

## 6 CONCLUSION

In this work, we introduce OJBench, a competition-level code reasoning benchmark. Our evaluation of 37 models reveals that even state-of-the-art reasoning-oriented models face significant challenges when tackling complex code reasoning tasks. Furthermore, we conducted in-depth experiments and analyses on the models based on this dataset, with the aim of providing valuable insights for future code LLM development.

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

## A    ETHICS STATEMENT

We ensure that we comply with applicable laws and ethical standards during the data collection and usage processes, and provide adequate compensation to all crowd workers. Since this benchmark involves objective knowledge and reasoning in the field of programming, the annotation content is not affected by geographical or cultural differences among annotators.

Furthermore, our dataset does not contain any personally identifiable information or offensive content. The authenticity and accuracy of OJBench have been thoroughly verified, providing a reliable basis for evaluating LLMs. OJBench is intended solely for academic and research purposes. Any commercial use or other misuse that deviates from this purpose is strictly prohibited. We will urge all users to respect this regulation in order to maintain the integrity and ethical use of this valuable resource.

## B    REPRODUCIBILITY STATEMENT

We are fully committed to the principles of reproducible research. To this end, we provide the following resources and information.

**Dataset.**  The complete OJBench benchmark, consisting of 232 competition-level problems, each with C++ and Python variants, will be made publicly accessible. This includes problem descriptions, canonical solutions, and test cases.

**Codebase.**  Our full evaluation framework will be open-sourced. This repository will contain all scripts required for executing the evaluation pipeline, from model querying to the final calculation of performance metrics.

**Experimental Details.**  As detailed in Section 3.1, this paper provides a comprehensive description of our experimental setup. This includes all model hyperparameters (e.g., temperature, top-p, max tokens), ensuring that our results can be precisely replicated.

## C    USE OF LARGE LANGUAGE MODELS

Large language models (LLMs) were employed solely as linguistic aids; their function was strictly limited to polishing pre-existing text, correcting grammar, and adjusting sentence structure so as to enhance readability and fluency. All core research content, innovative insights, experimental designs, data analyses, result interpretations, and scholarly viewpoints presented in this paper were developed independently by the authors. The LLMs were not used for research conceptualization, theoretical derivation, experimental planning, or the formulation of any principal conclusions.

## D    SUPPLEMENTARY DATASET

The complete benchmark dataset used in this research is provided as supplementary material in a single compressed file named OJBench_testdata.zip.

The dataset is organized as follows:

problems.jsonl: A JSON Lines file that contains detailed metadata for every problem, including descriptions, difficulty levels.

/ICPC/ and /NOI/ folders: These two folders contain the full set of test cases.

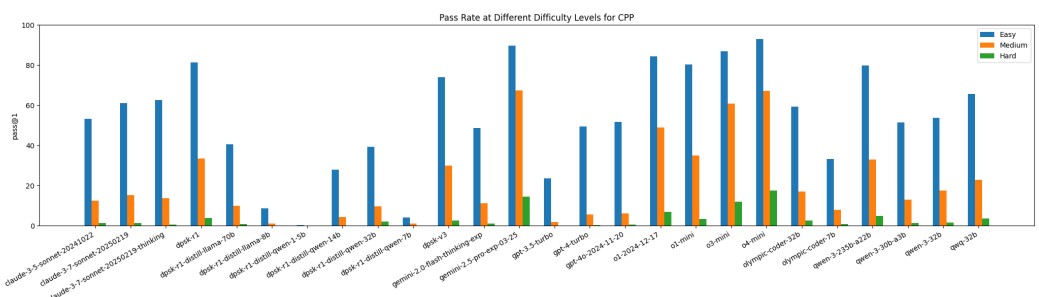

Figure 6: Pass rate at different difficulty levels of more models using CPP on OJBench

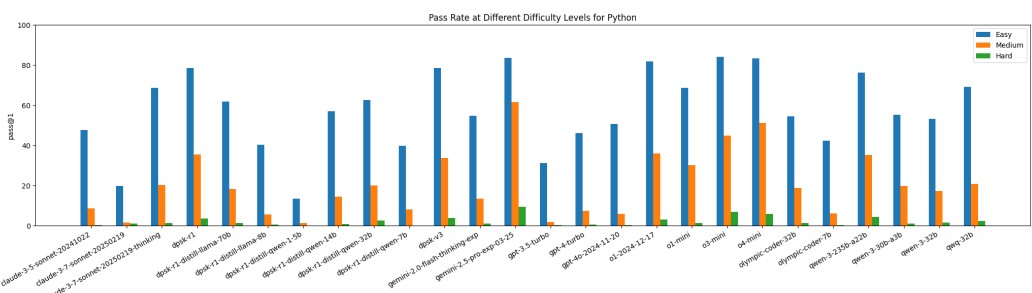

Figure 7: Pass rate at different difficulty levels of more models using Python on OJBench

## E    LIMITATIONS AND BROADER IMPACT

This paper introduces OJBench, which is used to evaluate the code reasoning abilities of LLMs and provides guidance for the research field. However, this paper still has some limitations.

**Limitations.** (1) Limited coverage: The data of OJBench mainly comes from open-source programming competition platforms and official open-source data from competitions, and it is of high difficulty. Although our benchmark is relatively more difficult compared to other benchmarks, given the breadth of the code specialization field, our evaluation cannot cover the entire scope of programming competitions. (2) Insufficient diversity: The problem types of OJBench are mainly focused on the field of algorithm competitions. For some emerging programming application scenarios, such as the development combining artificial intelligence and code, Internet of Things programming, and blockchain smart contract development, there is insufficient support for evaluating code reasoning abilities.

**Broader Impact.** In the research field, OJBench is expected to play an important role in the field of computer science. With the help of OJBench, researchers can accurately evaluate the performance of LLMs in human programming competitions, thereby promoting the development of scientific research.

## F    THE PASS RATES OF DIFFERENT DIFFICULTY LEVELS OF MODELS USING PYTHON AND CPP ON OJBENCH

Figure 6 and 7 show the pass rates of different difficulty levels in OJBench for more models using Python and CPP.

## G   DETAILS ABOUT THE INFERENCE EXPERIMENTS ON THE OPEN SOURCE MODELS

For inference experiments on all open source models with less than 72B parameters, we used two computing clusters equipped with 8 NVIDIA A100-80GB GPUs. Inference for each model took about two hours.

## H   DETAILS OF NOI AND ICPC

**NOI.** The National Olympiad in Informatics is one of the five major Olympiads in China. It is characterized by its high level of difficulty, encompassing a wide range of knowledge areas such as algorithms, data structures, combinatorial mathematics, and computational geometry. The problems in NOI are highly flexible, focusing not only on the contestants' grasp of fundamental computer science knowledge but also on their logical thinking abilities in solving complex problems, as well as the depth of their thinking and the breadth of their knowledge.

**ICPC.** The International Collegiate Programming Contest is one of the most influential collegiate computer programming contests globally. This competition is conducted in teams, with each team consisting of three participants who are required to solve a series of complex programming problems within a specified time frame. The organizational structure of the ICPC includes regional contests and a global final.

