# OpenReview forum: "OJBench: A Competition Level Code Benchmark For Large Language Models"
_ICLR.cc/2026/Conference — ICLR 2026 Conference Withdrawn Submission_

### Official Review · Reviewer_wvdd · 2025-10-16

**Soundness:** 3
**Presentation:** 3
**Contribution:** 3
**Rating:** 4
**Confidence:** 4

**Summary:**

This paper introduces OJBench, a competition-level code benchmark designed to evaluate LLMs on challenging programming problems. Built from 232 real problems sourced from NOI and ICPC, OJBench covers multiple difficulty levels (Easy/Medium/Hard), supports Python and C++, and uses a strict execution-based evaluation with full test cases. The benchmark assesses 37 models across open-/closed-source and general-/reasoning-oriented settings, revealing that even state-of-the-art models struggle on hard problems. The contributions include: (1) constructing the first large-scale benchmark with competition-grade coding tasks, (2) providing a fair, reproducible evaluation framework with pass@n metrics, and (3) offering detailed empirical analysis and insights into the limitations of current LLMs for competitive programming.

**Strengths:**

The paper demonstrates strong originality by introducing the first benchmark explicitly focused on competition-level programming tasks, bridging the gap between existing simple code benchmarks and the high-level reasoning challenges posed by contests like NOI and ICPC.

In terms of quality, the dataset construction is rigorous: problems are carefully curated, filtered, translated, and annotated with difficulty levels based on real contest data. The evaluation framework is robust, using execution-based full test cases and pass@n metrics, ensuring fairness and reproducibility across models.

The work has good clarity, with clear task categorization (Easy/Medium/Hard), transparent benchmark design, and insightful analyses including comparisons between Python and C++ performance, general vs. reasoning models, and case studies of model errors.

Finally, the significance is high: OJBench exposes a substantial performance gap between humans and state-of-the-art LLMs, showing that even advanced models struggle with hard competitive tasks. This establishes OJBench as a valuable resource for driving future research in code reasoning, model refinement, and reinforcement learning for programming.

**Weaknesses:**

Although this paper is very solid, there already exist many similar works. I hope the authors can highlight more clearly how OJBench differs from other competitive programming benchmarks such as LiveCodeBench, LiveCodeBench Pro, ACOBench, and CodeELO, and what specific advantages OJBench provides over them.

Since the level of novelty is somewhat limited, I would also encourage the authors to conduct more interesting analytical experiments that can yield unique insights. If OJBench can demonstrate the ability to provide distinctive insights compared to prior benchmarks, I would be inclined to raise my score.

**Questions:**

What is the most unique advantage of OJBench? At present, it seems unlikely that anyone would actually choose to work on this benchmark.

---

### Official Review · Reviewer_miMx · 2025-10-19

**Soundness:** 3
**Presentation:** 3
**Contribution:** 2
**Rating:** 2
**Confidence:** 4

**Summary:**

This paper introduces OJBench, a new benchmark designed to evaluate the code reasoning abilities of Large Language Models (LLMs) on competition-level problems. The authors argue that existing benchmarks like LiveCodeBench are becoming saturated and are not challenging enough for modern reasoning-focused LLMs.

OJBench consists of 232 problems sourced from China's National Olympiad in Informatics (NOI) and the International Collegiate Programming Contest (ICPC). The benchmark leverages the official test cases provided by the competition organizers. The authors evaluate an extensive set of 37 closed-source and open-source models, analyzing performance based on difficulty, programming language (Python vs. C++), and the models' ability to self-correct using error feedback. The main conclusion is that even state-of-the-art models struggle significantly with these problems, indicating a substantial gap in their advanced reasoning capabilities.

**Strengths:**

1. The selection of problems from NOI and ICPC is a clear strength, providing a high-difficulty set of tasks that effectively challenges the current generation of LLMs.

2. The evaluation of 37 different models is comprehensive. It provides a valuable, wide-ranging snapshot of the current landscape, from general-purpose coders to specialized reasoning models.

3. The inclusion of both C++ and Python evaluations is a thoughtful touch that reflects real-world competitive programming. Furthermore, the analysis of model refinement based on execution feedback (Section 4.3) offers practical insights into the iterative problem-solving capabilities of LLMs.

**Weaknesses:**

1. The most glaring omission is the absence of a data contamination analysis. Problems from high-profile competitions like NOI and ICPC are extensively discussed online, with countless solutions, tutorials, and analyses available on platforms like GitHub, blogs, and forums. It is almost certain that this data is present in the training corpora of the models being evaluated. Without a rigorous decontamination study to identify and potentially exclude contaminated problems, the benchmark's results are suspect. The reported performance may reflect memorization rather than true reasoning ability, which undermines the paper's primary goal.

2. The paper presents the use of "comprehensive test cases released by the competition organizers" as a strength. However, this is also a significant weakness. Simply collecting and using official test cases is not a novel methodological contribution. The value of a benchmark often lies in the curation, enhancement, and rigorous validation of its evaluation suite. Moreover, the paper assumes these official test cases are of high quality but provides no independent verification of their comprehensiveness. The only filtering step mentioned is discarding samples where an accepted solution fails the tests. This is a minimal sanity check, not a robust quality assessment. It does not guarantee that the test suite can catch subtle bugs or edge cases that differentiate good solutions from great ones.

3. The methodology for classifying problem difficulty is not well-substantiated and appears arbitrary. While the proposed classification methodology somehow makes sense, but it might fail to provide accurate insights, as this benchmark is targeting the top-tier programming problems.
- NOI Difficulty: Relying on user-voted difficulty ratings from a third-party platform is highly subjective and prone to noise. It is a weak proxy for objective difficulty.
- ICPC Difficulty: The authors invent a formula based on pass rates and attempt rates. While an interesting heuristic, it is not validated. This formula could be easily skewed by contest dynamics (e.g., teams making many trivial attempts on a hard problem) and is not a principled, reproducible measure of a problem's inherent complexity.

**Questions:**

Please see me pros and cons.

---

### Official Review · Reviewer_DUhW · 2025-11-01

**Soundness:** 1
**Presentation:** 1
**Contribution:** 1
**Rating:** 2
**Confidence:** 5

**Summary:**

This paper introduces OJBench, which collects 232 programming competition problems and evaluates 37 models. The experiments reveal limitations of current LLMs on competition-level coding and find that C++ is more effective than Python.

**Strengths:**

This is a complete paper: the data pipeline is effective, and the experiments are comprehensive. However, the weaknesses are, overall, quite serious.

**Weaknesses:**

- In line 047, the authors claim that CodeElo is not standardized and transparent, which are pivotal limitations of current coding benchmarks. In my view, this claim may not hold. A transparent benchmark can exacerbate data contamination, which is unacceptable. Moreover, if the authors argue that existing coding benchmarks may suffer evaluation bias due to the choice of problems, then OJBench would require strict quality control, diversity control, problem selection control, and robust data contamination safeguards. The paper’s current content does not demonstrate clear, significant differences from existing coding benchmarks along these dimensions.
- The paper should better articulate the novelty of OJBench. Compared to established benchmarks [1, 2, 3], it does not demonstrate significant advances in robustness or rigor. Furthermore, its task coverage is not broad enough to be considered a replacement for existing coding benchmarks. The authors need to clarify its distinct contributions.

*[1] McEval: Massively Multilingual Code Evaluation*

*[2] MdEval: Massively Multilingual Code Debugging*

*[3] BigCodeBench: The Next Generation of HumanEval*

**Questions:**

None.

---

### Official Review · Reviewer_8wpw · 2025-11-03

**Soundness:** 2
**Presentation:** 2
**Contribution:** 2
**Rating:** 4
**Confidence:** 4

**Summary:**

This paper introduces a new benchmark for evaluating code reasoning ability. The benchmark consists of 232 competition-level programming problems in NOI and ICPC. The data including the test cases is crawlered from Logu OJ. The authors benchmark 37 models (open- and closed-source) and show that even the strongest reasoning models struggle on hard problems. The study also finds that C++ performs better than Python, and iterative refinement using execution feedback can further improve accuracy.

**Strengths:**

1. This paper is easy to follow. It introduces the first fully competition-level code reasoning benchmark (OJBench) derived from real NOI and ICPC problems, addressing a clear gap beyond existing datasets like LiveCodeBench or CodeElo.
2. This paper shows how models can improve with execution-based refinement, providing insight into iterative reasoning and debugging behavior.
3. The authors plan to release dataset, codebase, and evaluation pipeline, ensuring transparency and long-term benchmark utility.

**Weaknesses:**

1. The analysis and conclusion are not deep; I did not gain new insights beyond the benchmark construction.
2. The paper mainly focuses on dataset creation and evaluation, which may fit better in a dataset or benchmark track rather than the main ICLR track.

**Questions:**

1. How do you ensure the fairness of OJBench when the problems come from two distinct sources (NOI and ICPC) with potentially different judging styles and difficulty calibration?
2. Could you elaborate on why C++ leads to consistently better results—do models actually reason differently, or is this mainly due to execution efficiency?
3. Have you analyzed whether model size or specific RL/distillation strategies correlate more strongly with performance on OJBench?
4. How do you mitigate potential data leakage, given that some NOI or ICPC problems (or similar variants) might already exist in public training corpora used by large models?

---

### Note · Authors · 2026-01-04

I have read and agree with the venue's withdrawal policy on behalf of myself and my co-authors.